# The Conopeptide αD-FrXXA, an Inhibitor of Voltage-Gated Potassium Channels

**DOI:** 10.3390/md23060237

**Published:** 2025-05-30

**Authors:** Luis Martínez-Hernández, Estuardo López-Vera, Ximena C. Rodriguez-Ruiz, Mónica A. Ortíz-Arellano

**Affiliations:** 1Posgrado en Ciencias Biológicas, Instituto de Ciencias del Mar y Limnología, Universidad Nacional Autónoma de México, Ciudad de México 04510, Mexico; lamh14@gmail.com; 2Laboratorio de Toxinología Marina, Unidad Académica de Ecología y Biodiversidad Acuática, Instituto de Ciencias del Mar y Limnología, Universidad Nacional Autónoma de México, Ciudad de México 04510, Mexico; ximena_crr@ciencias.unam.mx; 3Laboratorio de Malacología, Facultad de Ciencias del Mar, Universidad Autónoma de Sinaloa, Mazatlán 82000, Mexico; manabel@uas.edu.mx

**Keywords:** Kv channels, conotoxins, αD-superfamily, *Conus fergusoni*

## Abstract

The conopeptide αD-FrXXA was previously isolated by our team from the venom of the vermivorous snail *Conus fergusoni*. This toxin is composed of two chains of 47 amino acids and inhibits neuronal and muscular subtypes of nAChR. In this study, we explored its effects on voltage-gated potassium channels heterologously expressed in *Xenopus laevis* oocytes using the two-electrode voltage-clamp technique (TEVC). At a concentration of 15 μM, αD-FrXXA was able to inhibit by 50% or more the currents of four subtypes of the Kv1 subfamily and slightly inhibit (<20%) two subtypes of the EAG subfamily. The conopeptide αD-FrXXA inhibits in a concentration-dependent manner the subtypes Kv1.3 (IC_50_ 0.38 ± 0.06 μM) and Kv1.6 (IC_50_ 0.52 ± 0.14 μM). The results reported here are noteworthy because this α-conopeptide behaves similarly to the α/κJ-PlXIVA conopeptide that inhibits nAChR and Kv channels.

## 1. Introduction

*Conus* is a genus of marine snails that comprises approximately 1000 living species, and all of them are venomous predators [1]. These snails live in tropical and subtropical seas [2], where they are specialized predators that utilize a complex cocktail of venom peptides for defence and facilitate predation on worms, other molluscs, and small fishes [3].

The venom of cone snails contains a repertoire of several hundred different bioactive components [1]. Most of them are small peptides (conopeptides) ranging in size from 10 to 40 amino acids, even though there are cases of conopeptides that comprise over 40 amino acids [4]. Conopeptides can act on ligand-gated or voltage-gated ion channels with high potency and selectivity. The five best-studied classes of conopeptides target ion channels expressed in the nervous and locomotor systems: α-conopeptides (inhibitors of nicotinic acetylcholine receptors, nAChR), ω-conopeptides (inhibitors of voltage-gated calcium channels, Cav), κ-conopeptides (inhibitors of voltage-gated potassium channels, Kv), μ-conopeptides (inhibitors of voltage-gated sodium channels, Nav), and δ-conopeptides (delayers of activation of Nav) [5].

*Conus fergusoni* is a species that feeds on marine worms. Its distribution extends north from Bahía Tortuga, Baja California, and Guaymas, Sonora, in México to as far south as Isla Lobos in Peru [6]. The conopeptide αD-FrXXA is a neurotoxin from the venom of *Conus fergusoni* previously characterized by our team. This peptide is composed of two chains of 47 amino acids and is able to inhibit neuronal (hα7, hα4β2, hα3β2) and muscular (m(α1)_2_β1δε, m(α1)_2_β1δγ) nAChR [7].

As of August of 2024, seventeen toxins with activity acting on two groups of potassium channels have been reported in *Conus* venom [8], and only one, the α/κJ-PlXIVA conopeptide, can inhibit both nAChR and Kv channels [9]. In this article, we explore the possibility that αD-FrXXA also targets Kv. Surprisingly, results indicate that this conopeptide at a concentration of 15 µM can slightly inhibit (<20%) the currents of the Kv10.1 and Kv11.1 subtypes and inhibit the Kv1.3–Kv1.6 subtypes by more than 50%.

## 2. Results

### Activity of αD-FrXXA on Kv1 and EAG Subfamilies

To investigate the possible activity of αD-FrXXA on the currents of four subtypes of the Kv1 subfamily (Kv1.3–Kv1.6) and two of the EAG subfamily (Kv10.1, Kv11.1), initial tests were carried out with 15 μM of αD-FrXXA on potassium current (I_k_) amplitudes. At this concentration, a small blocking (<20%) of current was observed for Kv10.1 and Kv11.1 (Figure 1). On the other hand, the blocking effect was ≈50% for Kv1.4 and Kv1.5 and greater than 80% for Kv1.3 and Kv1.6 (Figure 1). It is important to note that the effect on Kv1.5, Kv10.1, and Kv11.1 was completely reversible in the first five minutes of washing, while for Kv1.3, Kv1.4, and Kv1.6, the effect was irreversible even for washings longer than sixty minutes (Figure 2a,b).

The concentration-response curves of αD-FrXXA activity on Kv1.3 and Kv1.6 is shown in Figure 3. The conopeptide αD-FrXXA had an IC_50_ of 0.38 ± 0.06 μM for Kv1.3 and an IC_50_ 0.52 ± 0.14 μM for Kv1.6. The irreversible effect was persistent at all concentrations tested on Kv1.3 and Kv1.6, as shown in Figure 4.

## 3. Discussion

Previously, Rodriguez-Ruiz et al. reported the purification and electrophysiological characterization of conopeptide αD-FrXXA from the venom of *C. fergusoni* [7]. This αD-conopeptide is a homodimeric toxin composed of two chains of 47 amino acids in length and inhibits the acetylcholine-induced response of nAChR with an IC_50_ of 125 nM on hα7, 282 nM on hα3β2, 697 nM on hα4β2, 351 nM on mouse adult muscle (m(α1)_2_β1δε), and 447 nM on mouse fetal muscle (m(α1)_2_β1δγ).

All reported conopeptides belonging to the D-superfamily are known to inhibit the nAChR and are longer than 40 amino acids (per chain) and able to form dimers (homo-, pseudohomo-, and heterodimers) [10,11]. Currently, eight αD-conopeptides (including αD-FrXXA) have been characterized from six *Conus* species, the affinity of five conopeptides have been determined on nAChR (αD-VxXXA, αD-VxXXB, and αD-VxXXC from *C. vexillium* [2]; αD-GeXXA from *C. generalis* [12]; and αD-PiXXA from *C. princeps* [13]), and two conopeptides, αD-MsXXA and αD-Cp20 from *C. mustelinus* and *C. capitaneus*, respectively, that present inhibitory activity but their affinity has not been studied in depth [10,14]. αD-conopeptides, unlike other α-conopeptides such as αA-RgIA (sometimes referred as “traditional” α-conopeptides), are characterized as allosteric modulators; in other words, they do not bind to acetylcholine-binding site [15].

In this work, we demonstrated that αD-FrXXA was able to inhibit the Kv1.3 and Kv1.6 subtypes at similar concentration values, IC_50_ 0.38 μM and 0.52 μM, respectively (Figure 3). However, the Hill coefficient value, which was greater than 1 (1.88 for Kv1.3 and 8.49 for Kv1.6), suggests that there was positive cooperativity to inhibit the channels [16], indicating more than one binding site, especially for the Kv1.6 subtype.

Molecular dynamics (MD) simulations applied to protein–ligand complexes have proposed three interaction pathways between conopeptides and potassium channels. The first takes place when the toxin blocks the pore of the channel by a functional dyad composed of a positively charged amino acid and an aromatic amino acid [17], a mechanism known as “molecular plug”; an example of this is κO-PVIIA [18,19]. In the second type of interaction, the conopeptide triggers the collapse of the selectivity filter by targeting channel residues that gate the access of water molecules to the aqueous cavities around the pore (“porecollapse”); this type of effect was reported for Conkunitzin-S1 [20]. And the third refers to when the toxin interacts with the channel in a manner to divert the permeant ions and trap them in off-axis cryptic sites above the selectivity filter, a mechanism termed as a “molecular lid”; this last effect was reported for Conkunitzin-C3 [21]. Taking the latter into account, the αD-FrXXA conopeptide, being even larger than Conkunitzin-S1 and Conkuniztin-C3 (60 amino acid), may have more than one interaction pathway with Kv1.3 and Kv1.6. However, further studies are needed to understand the mechanism of action of this conopeptide.

The αD-FrXXA conopeptide is the second component of *Conus* venom found to target nAChR and Kv (47 amino acids, including 10 cysteine residues for each chain) and, like α/κJ-PlXIVA from *C. planorbis* (25 amino acids in length, including 4 cysteine residues), shares the molecular targets m(α1)_2_β1δε and Kv1.6, and αD-FrXXA has a higher affinity for both subtypes: 351 vs. 540 nM for m(α1)_2_β1δε, and 0.52 vs. 1.59 μM for Kv1.6 [7,9]. Moreover, the fact that the venom of vermivorous species contains toxins capable of inhibiting two molecular targets could be of ecological importance, especially because of the concentration at which they exert their activity.

Studies on the venom of piscivorous *C. purpurascens* have identified conopeptides involved in paralysis processes during prey hunting. The synergistic action of several conopeptides has been described to produce paralysis, a mechanism called “toxin cabal” [22]. The first physiological effect is called “lightning-strike cabal”, refers to the rapid immobilization of prey [22], and is mainly due to the action of the conopeptides δO-PVIA and κO-PVIIA, which have Nav (Nav1.2, Nav1.4 and Nav1.7) and Kv (*Shaker*), respectively, as molecular targets [17,23]. The second effect is the total inhibition of neuromuscular transmission [22], the “motor cabal”. This effect is due to the αA-PIVA and μM-PIIIA conopeptides, which have been reported to target nAChR m(α1)_2_β1δε and m(α1)_2_β1δγ and Nav1.2-Nav1.8 [24,25], respectively. Furthermore, μM-PIIIA at 10 μM has been reported to inhibit Kv1.6 currents by 80.6% [26]. In view of this information, we found that the αD-FrXXA conopeptide shares molecular targets with some conopeptides involved in the paralysis process by *C. purpurascens* venom, which could be an indication of a similar capture mechanism in the vermivorous species *C. fergusoni*. However, this theory is still questionable due to the lack of data on the distribution of these ion channel subtypes in marine worms and the need for behavioural bioassays similar to those carried out by Terlau and Olivera [22].

Seventeen *Conus* toxins with activity on potassium channels have been reported [8]. In this regard, the αD-FrXXA conopeptide ranks 18th in potassium channel activity and 10th *Conus* toxin that targets the Kv1.6 subtype, making it the most common potassium channel subtype targeted by conopeptides. Therefore, it is not an outlandish idea to think that the Kv1.6 subtype could be ecologically relevant, and perhaps the same could be true for the Kv1.3 subtype.

Finally, as the αD-FrXXA conopeptide is a structurally very large toxin, its possible activity on other ion channels, e.g., Nav and even Cav, remains to be explored. This would broaden the affinity spectrum of the D-superfamily of conopeptides, which would suggest that a single toxin from a vermivorous *Conus* could be sufficient during prey hunting.

## 4. Materials and Methods

### 4.1. Isolation of αD-FrXXA from Crude Venom Extract of Conus fergusoni

The isolation of αD-FrXXA from the venom of *C. fergusoni* was performed as originally described by Rodriguez-Ruiz et al. [7].

### 4.2. Preparation of Vectors Encoding Potassium Channels

cDNA of the clones for human Kv1.3, Kv1.4, Kv1.5, Kv1.6, Kv10.1, and Kv11.1 channels in vector pSGEM, resistant to ampicillin, were linearized with NotI, EcoRI, NheI, SpHI, Sfil, and EcoRI, respectively. The products were purified using the *EZ-10 Spin Column PCR Purification Kit* (Bio Basic; Amherst, New York, NY, USA). cDNA was transcribed in vitro with T7 polymerase (mMessage mMachine Kit; Applied Biosystem, Foster City, CA, USA). cRNA was purified using a *Qiagen RNeasy Kit* (QIAGEN; Redwood City, CA, USA).

### 4.3. Expression of Voltage-Gated Potassium Channels

Stage V-VI oocytes were isolated by microsurgery of anesthetised *Xenopus laevis* frogs. Sexually mature *X. laevis* female frogs were kept and handled according to the institutional bioethics committee requirements. The anesthetised process consisted of 30 min submersion in 2% tricaine methane sulfonate (MS-222; Merck KGaA, Darmstadt, Germany) before microsurgery was performed. Isolated oocytes were defolliculated by treatment with 0.75 mg/mL collagenase A (Roche; San Jose, CA, USA; Cat. #10103586001) and kept in ND96 solution (96 mM NaCl, 2 mM KCl, 1.8 mM CaCl_2_·2H_2_O, 1 mM MgCl_2_·6H_2_O, 5 mM HEPES, pH 7.3) supplemented with penicillin/streptomycin at (100 U/100 µg)/mL and 100 µg/mL gentamicin at 15 °C.

For heterologous expression, individual oocytes were injected with 1.3 ng of Kv1.3, 18.4 ng of Kv1.4 and Kv1.6, and 9.2 ng of Kv1.5, Kv10.1, and Kv11.1 using a micro-injector, Nanoliter 2000 (World Precision Instruments; Sarasota, FL, USA).

### 4.4. Electrophysiological Recordings

Ionic currents were measured at room temperature (20–22 °C) using an OC-725C clamp amplifier (Warner Instruments; Holliston, MA, USA) between 8 and 72 h after injections of cDNA using the two-electrode voltage-clamp technique (TEVC). Data acquisition was performed with the programme *LabView* (National Instruments, Chihuahua, Mexico). Standard borosilicate microelectrodes used for recording were filled with 3 M KCl solution and had a resistance of 0.5–1 MΩ. The bath solution was ND96, consisting of 96 mM NaCl, 2 mM KCl, 1.8 CaCl_2_, 1 mM MgCl_2_, and 5 mM HEPES (pH7.2) with pen/strep at (100 U/100 µg)/mL and 100 µg/mL of gentamicin. To assay conopeptide αD-FrXXA, we stopped the flow and directly applied it (diluted in DN96, 3 µL) to the bath chamber (30 µL); once maximum inhibition was reached at a given concentration, the flow for conopeptides washout was restarted. The duration of records was 1300 ms for Kv1.3–Kv10.1 and 2600 ms for Kv11.1. Potassium currents were evoked by test pulses to +10 mV from a holding potential of −80 mV applied every 10 s. In order to analyze the effect of αD-FrXXA [5 μM] on Kv1.3 and Kv1.6, currents were elicited by test pulses that varied in 10 mV increments (every 10 s) from −70 to +80 mV from a holding potential of −80 mV.

### 4.5. Data Analysis

Measurements were reported as the mean ± standard error of the mean (SEM) of at least three independent experiments. The current inhibition % was calculated from the current amplitude values obtained with ND96 extracellular solution (I_Control_) and those of each of the experimental conditions (I_[αD-FrXXA]_); therefore, current inhibition % = [1 − (I_[αD-FrXXA]_/I_Control_)] × 100. Currents plots were made with SigmaPlot version 10 (Systat Software, Inc., San Jose, CA, USA). The construction of the concentration–response curve and bar graphs and determination of the mean inhibitory concentration (IC_50_) were carried out from the data of the current inhibition % as a function of the concentration (µM) of αD-FrXXA expressed in logarithmic form (log[αD-FrXXA]) using GraphPad Prism version 9.5.1 (GraphPad Software, San Diego, CA, USA). Concentration–response curves were fitted to the following equation: current inhibition % = 100/{1 + (log[αD-FrXXA]/IC_50_)*^n^*^H^}, where *n*H is the Hill coefficient.

## 5. Conclusions

In summary, the αD-FrXXA conopeptide is the second *Conus* venom conopeptide reported to inhibit nAChR and Kv, showing a higher affinity for the Kv1.3 and Kv1.6 subtypes than the Kv1.4, Kv1.5, Kv10.1, and Kv11.1 subtypes. Therefore, we concluded based on the advances reported here that this conopeptide should be renamed α/κD-FrXXA, similar to α/κJ-PlXIVA.

## Figures and Tables

**Figure 1 marinedrugs-23-00237-f001:**
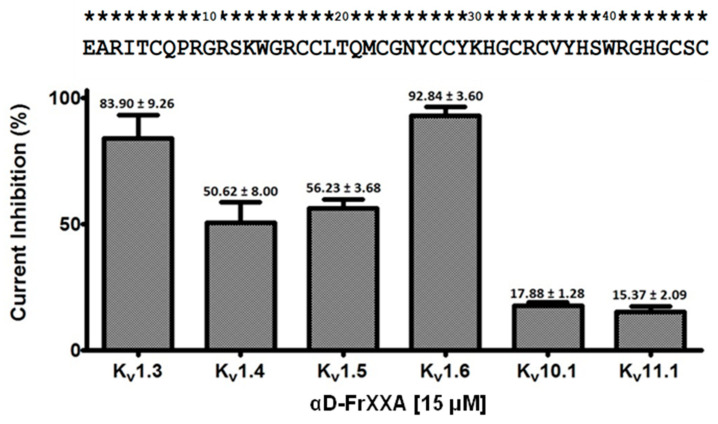
An overview of the inhibition of six Kv channels by αD-FrXXA at a concentration of 15 µM. The current inhibition (%) observed after the addition of 15 µM αD-FrXXA to various channels is displayed. Values are shown as the mean ± S.E.M. of at least three independent experiments (*n* ≥ 3). The sequence of amino acid residues of one of the chains of the αD-FrXXA homodimer is shown in the upper part of the figure. The standard one-letter code is employed for amino acids.

**Figure 2 marinedrugs-23-00237-f002:**
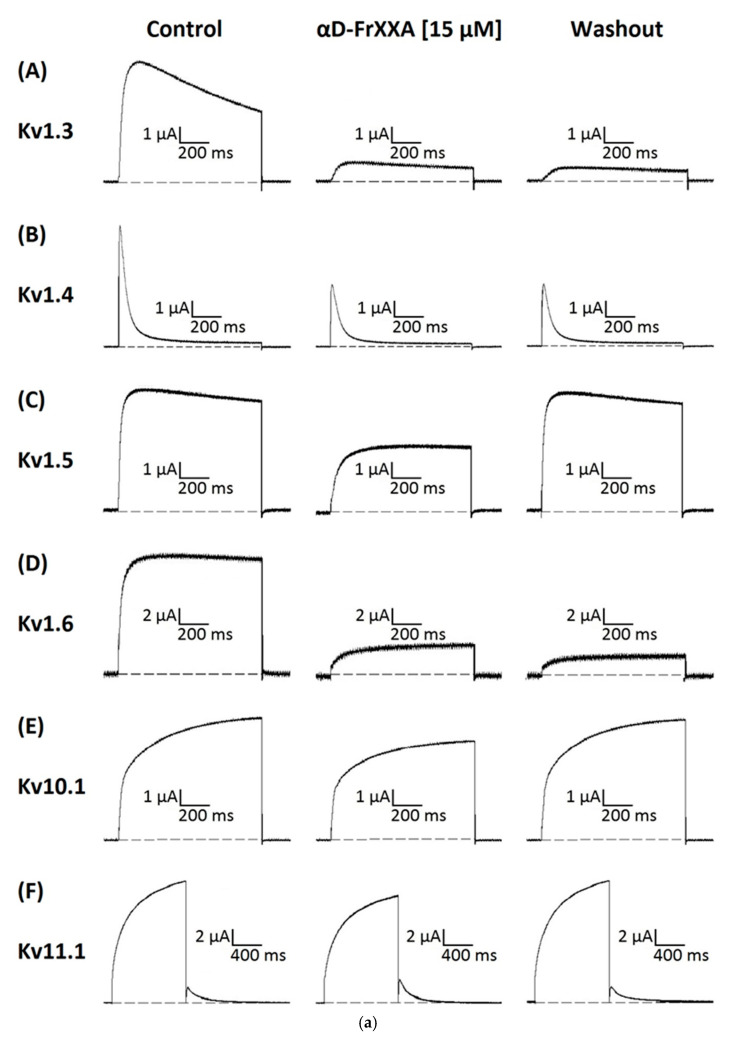
(**a**) Inhibition of six subtypes of Kv channels by αD-FrXXA. Representative recording showing control currents of Kv1.3 (**A**), Kv1.4 (**B**), Kv1.5 (**C**), Kv1.6 (**D**), Kv10.1 (**E**), and Kv11.1 (**F**) (left panel), in presence of 15 μM of αD-FrXXA (middle panel) and after 5 min (Kv1.5, Kv10.1, and Kv11.1) or 60 min of washout (Kv1.3, Kv1.4 and Kv1.6) (right panel). Currents were evoked by depolarising pulses to +10 mV from holding potential of −80 mV applied every 10 s.(**b**) Time course of inhibition by αD-FrXXA at 15 μM (static flow) from normalized current (I/Imax) on six subtypes of Kv channels.

**Figure 3 marinedrugs-23-00237-f003:**
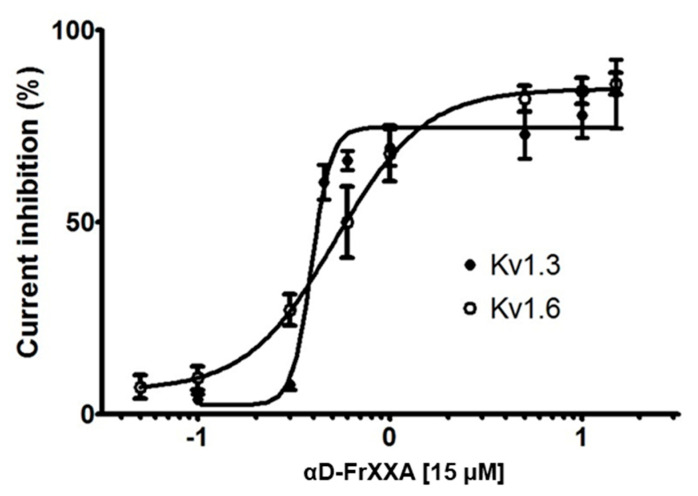
The fitting of a Hill equation to concentration–response curves for the inhibition of Kv1.3 and Kv1.6. Concentration-response curves are shown as a plot of the percentage of blocked current as a function of increasing toxin concentration. Closed circles represent Kv1.3 (IC_50_ 0.38 ± 0.06 μM, Hill coefficient 8.49 ± 3.23) and open circles represent Kv1.6 (IC_50_ 0.52 ± 0.14 μM, Hill coefficient 1.88 ± 0.52). Each point corresponds to the mean ± S.E.M. of at least three independent experiments (*n* ≥ 3).

**Figure 4 marinedrugs-23-00237-f004:**
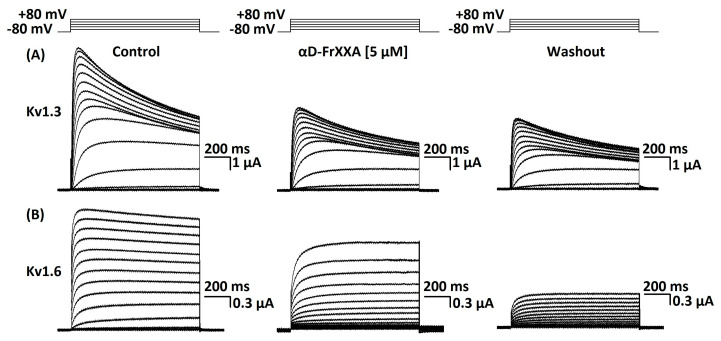
Representative traces obtained from two Kv channels: Kv1.3 (**A**) and Kv1.6 (**B**). Currents were evoked using a protocol schematized in the upper part of the figure (see details in the text), in the control condition (left panel), during the addition of 5 µM of αD-FrXXA (middle panel), and after 60 min of washout (right panel). The traces shown are representative of at least three independent experiments (*n* ≥ 3).

## Data Availability

The original contributions presented in this study are included in the article. Further inquiries can be directed to the corresponding author.

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
