# Peer review of "The Conopeptide αD-FrXXA, an Inhibitor of Voltage-Gated Potassium Channels"

_marinedrugs, 2025, doi:10.3390/md23060237_

Round 1

Reviewer 1 Report

Comments and Suggestions for Authors

This manuscript presents interesting electrophysiological data, demonstrating for the first time that αD-FrXXA, previously identified as an nAChR antagonist, also inhibits voltage-gated potassium (Kv) channels. The data is interesting; however, several points must be addressed to improve the clarity, accuracy, and interpretability of the findings:

Major Comments:

  1. Residual Instructional Text:
    • The manuscript contains what appears to be leftover text from the journal’s instructions to authors ("This section may be divided by subheadings..."). Please remove this to maintain professional clarity.
  2. Figure Issues:
    • Figure 2: The lettering (A, B, C...) is misaligned. Please correct the figure alignment.
    • Figure 3: Clearly label the panels ("Kv1.3", "Kv1.6") directly within the figure panels, rather than only in the legend.
  3. Hill Coefficient and Channel Expression:
    • Kv channel sensitivity to toxins and blocking ions (such as Ba²⁺) depends strongly on expression density and overall current amplitude. The remarkably high Hill coefficient observed for Kv1.3 (8.49 ± 3.23) likely results from excessively large Kv1.3 currents, as inferred from the raw traces provided in the figures. In contrast, Kv1.6 expression appears inherently low, as indicated by the necessity for significantly higher cRNA injection (18.3 ng vs. only 1.3 ng for Kv1.3). To address this issue, I suggest substantially reducing the Kv1.3 cRNA injection amount (e.g., to approximately 0.2 ng) and recording currents at an earlier time point (24–48 hrs post-injection), ensuring the currents at +10 mV remain below approximately 5 µA. These conditions will likely normalize the Hill coefficient, allowing for more reliable affinity measurements
  4. Sequence Accessibility:
    • The mature-chain sequences of the toxin appear only as images in the authors' previous publication (Rodríguez-Ruiz et al., 2022). For clarity and reproducibility, please provide the exact 47-residue sequences explicitly either in a Supplementary Table or by referencing their ConoServer IDs.
  5. Quantifying Irreversibility:
    • The irreversibility of toxin effects in Figure 2 should be quantitatively demonstrated. I recommend adding a straightforward current amplitude (I/I₀) versus time plot clearly depicting baseline, toxin application, and wash phases for both reversible (e.g., Kv1.5) and irreversible effects (e.g., Kv1.3). This would substantially clarify the nature of the toxin-channel interaction.
  6. Preliminary Structural/Mechanistic Insights:
    • Given this is the second publication on αD-FrXXA by this group, incorporating a preliminary mechanistic analysis would significantly enhance the manuscript. Specifically:
      • Conduct TEA competition assays to clarify the site of action (vestibular or elsewhere).
      • Perform voltage-dependence assays to determine if toxin binding is sensitive to membrane voltage. Additionally, test if toxin binding can be reversed by strong depolarizing prepulses.
  • Using publicly accessible AlphaFold resources, I generated a structural model of αD-FrXXA bound to Kv1.3 in a few minutes. The model clearly suggests the toxin dimer sits atop the channel pore, positioning Arg9 near the channel’s selectivity filter (S0 site). Including a preliminary computational model (AlphaFold) combined with simple electrophysiological tests proposed above (TEA competition and voltage-dependence) would substantially increase the paper's mechanistic insight and provide a robust foundation for future experiments.

Recommendation:

Overall, the manuscript represents valuable research worthy of publication following revision.

Author Response

Major Comments:

  1. Residual Instructional Text:
    • The manuscript contains what appears to be leftover text from the journal’s instructions to authors ("This section may be divided by subheadings..."). Please remove this to maintain professional clarity.

Answer: Thank you very much; we have deleted the residual instructions.

      2. Figure Issues:

    • Figure 2: The lettering (A, B, C...) is misaligned. Please correct the figure alignment.

Answer: The lettering (A, B, C, …) changed at the moment to upload the document. We have done the corrections.

    • Figure 3: Clearly label the panels ("Kv1.3", "Kv1.6") directly within the figure panels, rather than only in the legend.

Answer: Thank you very much for the comment and we have modified Figure 4, as we believe that is the one you are referring to.

      3. Hill Coefficient and Channel Expression:

    • Kv channel sensitivity to toxins and blocking ions (such as Ba²⁺) depends strongly on expression density and overall current amplitude. The remarkably high Hill coefficient observed for Kv1.3 (8.49 ± 3.23) likely results from excessively large Kv1.3 currents, as inferred from the raw traces provided in the figures. In contrast, Kv1.6 expression appears inherently low, as indicated by the necessity for significantly higher cRNA injection (18.3 ng vs. only 1.3 ng for Kv1.3). To address this issue, I suggest substantially reducing the Kv1.3 cRNA injection amount (e.g., to approximately 0.2 ng) and recording currents at an earlier time point (24–48 hrs post-injection), ensuring the currents at +10 mV remain below approximately 5 µA. These conditions will likely normalize the Hill coefficient, allowing for more reliable affinity measurements

Answer: Thank you for the suggestion; Kv1.3 is relativity easy to express, reason that we use less amount of cRNA compared to the other Kv subtypes. However, when we tried to express Kv1.3 with 0.2 ng as you suggested, or even a little more (0.5ng), the oocytes express the channels 5 or 7 days after injection and the membrane can not maintain the voltage. We performed this experiment with oocytes from 3 donor frogs, and we observed the same problem. Nevertheless, we are still working on it to get a better Hill coefficient, but we do not know how long it will take.  We appreciate the recommendation for future experiments.

     4. Sequence Accessibility:

    • The mature-chain sequences of the toxin appear only as images in the authors' previous publication (Rodríguez-Ruiz et al., 2022). For clarity and reproducibility, please provide the exact 47-residue sequences explicitly either in a Supplementary Table or by referencing their ConoServer IDs.

Answer: We added on top of the figure1 the sequence of amino acids residues of one of the chains of the αD-FrXXA homodimer.

    5. Quantifying Irreversibility:

    • The irreversibility of toxin effects in Figure 2 should be quantitatively demonstrated. I recommend adding a straightforward current amplitude (I/I₀) versus time plot clearly depicting baseline, toxin application, and wash phases for both reversible (e.g., Kv1.5) and irreversible effects (e.g., Kv1.3). This would substantially clarify the nature of the toxin-channel interaction.

Answer: We added a new Figure named 2b, showing time course of inhibition by αD-FrXXA at 15 μM from the normalized current (I/Imax) on the six Kv channels. Thank you for the recommendation.

    6. Preliminary Structural/Mechanistic Insights:

    • Given this is the second publication on αD-FrXXA by this group, incorporating a preliminary mechanistic analysis would significantly enhance the manuscript. Specifically:
      • Conduct TEA competition assays to clarify the site of action  
      • Perform voltage-dependence assays to determine if toxin binding is sensitive to membrane voltage. Additionally, test if toxin binding can be reversed by strong depolarizing prepulses.
  • Using publicly accessible AlphaFold resources, I generated a structural model of αD-FrXXA bound to Kv1.3 in a few minutes. The model clearly suggests the toxin dimer sits atop the channel pore, positioning Arg9 near the channel’s selectivity filter (S0 site). Including a preliminary computational model (AlphaFold) combined with simple electrophysiological tests proposed above (TEA competition and voltage-dependence) would substantially increase the paper's mechanistic insight and provide a robust foundation for future experiments.

Answer: We greatly appreciate your recommendation. In this work, we sought to determine the potential activity of αD-FrXXA on potassium channels based on other K-conotoxins, such as KJ-PIXIVA, a preferential inhibitor of the Kv1.6 channel and also an inhibitor of the rat α3β4 and mouse α1β1eδ nicotinic acetylcholine receptors. The findings support our idea, and we agree with yours as to whether it acts in the pore of Kv channel or if it also competes for the acetylcholine binding site in the α7 nicotinic receptor (previous results), since it presents an irreversible inhibition for both membrane proteins. Nevertheless, we are willing to conduct the experiments; However, we only have few micrograms of αD-FrXXA available for this, and since it is a dimer, it is difficult to obtain by chemical synthesis.

Reviewer 2 Report

Comments and Suggestions for Authors

Three years ago (ref.7)  the authors have isolated a new conotoxin alpha-D-FrXXA and demonstrated  that it can inhibit muscle-  and  several neuronal  subtypes of nicotinic acetylcholine receptor (nAChR). In the submitted  manuscript they give a reference  of 2006 where a conotoxin alpha/rJ -P1XIV A which was shown to block not only some nAChRs , but also different Kv channels – apparently this cited work stimulated the authors to test the capacity of alpha-D-FrXXA  (earlier shown to block various nAChRs)  also to block some  Kv channels. The most efficient inhibition was observed  for the Kv 1.3 and 1.6 channels. For these two-types they determined IC50 values (which should be included into the Abstract).

The authors  mentioned that alpha-D-FrXXA consists  of  two chains of  47 amino acids, but say nothing about its spatial structure or a chemical or spatial structure of  alpha/rJ -P1XIV A which was the first to have inhibitory activity both against Kv channels and nAChRs.

 Finding of differences between Kv subtypes in the reversibility of blocking is of interest.

Minor comments:

Line 103 should be: five conopeptides have  well-characterized

Line 134: no need to give similar values once as nM and then as microM

Line 155 : no need to give several times references “  as  of August 2024”

Line 169 and 176: there is no need to describe these two experimental procedures as two sub-chapters

Discussion should include what is similar in the chemical and spatial (if available) structures  of the alpha-D-FrXXA and alpha/rJ -P1XIV A in order to assess the novelty and value of the presented results.

Graphical Abstract is mentioned, but I did not see it

Comments on the Quality of English Language

I noticed several minor mistakes, but in my view the English is in general acceptable

Author Response

Comments and Suggestions for Authors

Three years ago (ref.7)  the authors have isolated a new conotoxin alpha-D-FrXXA and demonstrated  that it can inhibit muscle-  and  several neuronal  subtypes of nicotinic acetylcholine receptor (nAChR). In the submitted  manuscript they give a reference  of 2006 where a conotoxin alpha/rJ -P1XIV A which was shown to block not only some nAChRs , but also different Kv channels – apparently this cited work stimulated the authors to test the capacity of alpha-D-FrXXA  (earlier shown to block various nAChRs)  also to block some  Kv channels. The most efficient inhibition was observed  for the Kv 1.3 and 1.6 channels. For these two-types they determined IC50 values (which should be included into the Abstract).

Answer: Thank you for your kind comment on the manuscript. We reviewed the abstract, and the IC50 values for Kv1.3 and Kv1.6 are included in the abstract.

The authors  mentioned that alpha-D-FrXXA consists  of  two chains of  47 amino acids, but say nothing about its spatial structure or a chemical or spatial structure of  alpha/rJ -P1XIV A which was the first to have inhibitory activity both against Kv channels and nAChRs.

 Finding of differences between Kv subtypes in the reversibility of blocking is of interest.

Minor comments:

Line 103 should be: five conopeptides have  well-characterized

Line 134: no need to give similar values once as nM and then as microM

Line 155 : no need to give several times references “  as  of August 2024”

Line 169 and 176: there is no need to describe these two experimental procedures as two sub-chapters

 Answer: Thank you so much; We have reviewed sentences throughout the manuscript and made changes based on your observation. In the case of two experimental procedures (Lines 169 and 176), we leave it as is, based on recommendations from other works and thus making it easier for readers to search for information.

Discussion should include what is similar in the chemical and spatial (if available) structures of the alpha-D-FrXXA and alpha/rJ -P1XIV A in order to assess the novelty and value of the presented results.

 Answer: Thank you very much again for your comment; we did not mention anything about the structural similarities between the two peptides because they are completely different peptides. In this case, αD-FrXXA is a homodimeric peptide with 47 aa of which 10 are cysteine residues for each chain meanwhile α/κJ-PIXIVA is a peptide composed of 25 aa in length, including 4 cysteine residues. However, we added above Figure1, the sequence of amino acids residues of one of the chains of the αD-FrXXA homodimer.

Graphical Abstract is mentioned, but I did not see it

Answer: We have improved the quality of the Graphical Abstract. Now, you can see it.

Round 2

Reviewer 1 Report

Comments and Suggestions for Authors

The authors have made significant improvements to the figures and data presentation but have not addressed the more substantive experimental suggestions regarding channel expression optimization and mechanistic studies.

Author Response

Comments and Suggestions for Authors

The authors have made significant improvements to the figures and data presentation but have not addressed the more substantive experimental suggestions regarding channel expression optimization and mechanistic studies.

Answer: We really appreciate your recommendation, and we apologize for having not been clear about the situation that we cannot perform more experiments to clarify the action mechanism of αD-FrXXA on Kv channels. In this moment, we are concerned, that we have less than 1nmol of the peptide with a molecular weight of 11,023.70 Da and to achieve lower Kv1.3 expression (up to 7 days after injection), oocytes die upon initiation of the depolarization protocol. Therefore, we need to collect more snails to generate more material (αD-FrXXA) and conduct the suggested experiments and perhaps buy a new batch of frogs to improve the expression.

Reviewer 2 Report

Comments and Suggestions for Authors

Since alpha/rJ -P1XIV A  was the first conotoxin known  block not only some nAChRs , but also different Kv channels ,  at least its chemical structure (chain length and  the  number of disulfide bonds should be mentioned in the manuscript.

Author Response

Comments and Suggestions for Authors

Since alpha/rJ -P1XIV A  was the first conotoxin known  block not only some nAChRs , but also different Kv channels ,  at least its chemical structure (chain length and  the  number of disulfide bonds should be mentioned in the manuscript.

Answer: Thank you very much for your comment. In the discussion section, we added the length, and cysteine residues for both peptides (αD-FrXXA and α/ĸJ-P1XIVA)